# *APOE* Locus-Associated Mitochondrial Function and Its Implication in Alzheimer’s Disease and Aging

**DOI:** 10.3390/ijms241310440

**Published:** 2023-06-21

**Authors:** Eun-Gyung Lee, Lesley Leong, Sunny Chen, Jessica Tulloch, Chang-En Yu

**Affiliations:** 1Geriatric Research, Education, and Clinical Center, VA Puget Sound Health Care System, Seattle, WA 98108, USA; lesley.leong@va.gov (L.L.); sunny.chen@va.gov (S.C.); jessica.tulloch@va.gov (J.T.); 2Department of Medicine, University of Washington, Seattle, WA 98195, USA

**Keywords:** *APOE* locus, *TOMM40*, *APOE*, oxidative stress, mitochondrial dysfunction, mitochondrial DNA copy number, gene expression, Alzheimer’s disease, aging

## Abstract

The Apolipoprotein E (*APOE)* locus has garnered significant clinical interest because of its association with Alzheimer’s disease (AD) and longevity. This genetic association appears across multiple genes in the *APOE* locus. Despite the apparent differences between AD and longevity, both conditions share a commonality of aging-related changes in mitochondrial function. This commonality is likely due to accumulative biological effects partly exerted by the *APOE* locus. In this study, we investigated changes in mitochondrial structure/function-related markers using oxidative stress-induced human cellular models and postmortem brains (PMBs) from individuals with AD and normal controls. Our results reveal a range of expressional alterations, either upregulated or downregulated, in these genes in response to oxidative stress. In contrast, we consistently observed an upregulation of multiple *APOE* locus genes in all cellular models and AD PMBs. Additionally, the effects of AD status on mitochondrial DNA copy number (mtDNA CN) varied depending on *APOE* genotype. Our findings imply a potential coregulation of *APOE* locus genes possibly occurring within the same topologically associating domain (TAD) of the 3D chromosome conformation. The coordinated expression of *APOE* locus genes could impact mitochondrial function, contributing to the development of AD or longevity. Our study underscores the significant role of the *APOE* locus in modulating mitochondrial function and provides valuable insights into the underlying mechanisms of AD and aging, emphasizing the importance of this locus in clinical research.

## 1. Introduction

The association between the *APOE* locus and AD is well established [1,2], and numerous studies have also linked this locus to longevity across diverse ethnic groups [3,4,5,6,7]. The genetic signals associated with AD risk or longevity in this locus cluster around four functionally distinct genes: *APOE*, *APOC1*, *NECTIN2*, and *TOMM40*. This region has also been shown to contain complex genomic structures encompassing multiple regulatory elements that impact various physiologies, including cognitive health, lipid metabolism, and immunity [8,9,10,11]. Due to reduced recombination, the four genes are in strong linkage disequilibrium with each other [12,13], and genetic signals are indistinguishable between surrogates and true effectors. While *APOE* has traditionally been considered the sole effector of AD risk in this locus, the involvement of the other three genes cannot be easily dismissed. Although AD and longevity may appear to be distinct clinical manifestations, they both share the commonality of aging. AD is an aging-associated neurodegeneration, whereas longevity can be considered successful aging. One of the prominent biological features of aging is mitochondrial dysfunction, which has been extensively studied in both AD and aging [14,15,16,17]. Therefore, mitochondrial function may serve as a focal point for deciphering the true biological effects of the *APOE* locus and understanding the molecular basis of AD and longevity.

Mitochondria are dynamic organelles that constantly reshape their networks within the cell [18], influencing mitochondrial function and modifying physiology [17]. One of their distinct features is the oxidative stress response. Cells overproducing or failing to remove oxygen-containing free radicals, also known as reactive oxygen species (ROS), can cause oxidative stress. Excess ROS can damage DNA, RNA, lipids, and proteins, ultimately reducing cells’ and tissues’ performance and function and interfering with self-organizing systems and their ability to adapt to the environment [19]. Lifestyle and environment also have strong impacts on the risk of dementia. In recent years, concerns have been raised regarding the potential health effects of prolonged exposure to microwave radiation emitted by electronic devices, such as smartphones, Wi-Fi routers, and microwave ovens. Both radio waves and microwaves fall within the non-ionizing region of the electromagnetic spectrum, which is characterized by relatively low energy levels. However, when absorbed into biological tissues, it can induce the production of ROS and free radicals, lead to oxidative stress, disrupt cellular homeostasis, and damage DNA, proteins, and lipids [20,21]. Mitochondria play a crucial role in managing oxidative stress, as they are responsible for energy production, metabolic regulation, ion homeostasis, and complex signaling pathways (such as stress response, immunity, and cell death). As mitochondria are the primary site for ROS generation during ATP production [22,23], defective mitochondria can become a source of excess ROS, triggering mitochondrial dysfunctions and leading to a vicious cycle [24]. Defects in mitochondrial function and dynamics, such as alterations in components, morphology, membrane potential, and mtDNA CN, can accelerate aging and contribute to AD risk [25,26,27,28,29].

The *APOE* locus has the potential to play a significant role in mitochondrial function. Among the four genes within the *APOE* locus, *TOMM40* has the most direct link with mitochondria. The majority of the estimated 1500 mitochondrial proteins are encoded by nuclear genes, and these proteins are synthesized by cytosolic ribosomes and transported via the translocase of the outer membrane (TOM) complex [30]. *TOMM40* encodes the TOM40 protein that creates a hydrophilic and cation-selective translocation pore of the TOM complex [31,32], functioning as a significant gateway for proteins to enter the mitochondrion [33]. The depletion of TOM40 may lead to increased levels of ROS, decreased mitochondrial integrity and ATP production, oxidative DNA damage, and mtDNA deletions [34]. The disruption of mitochondrial protein import due to the Aβ blockage of TOM40 has also been proposed as a trigger for mitochondrial dysfunction in AD [35,36]. In addition to *TOMM40*, there is substantial evidence linking *APOE* with mitochondrial function. PMBs from young *APOE* ε4 carriers showed reduced cytochrome oxidase activity [37,38]. Fluorodeoxyglucose positron emission tomography studies showed decreased glucose utilization in *APOE* ε4 carriers, indicating mitochondria-related bioenergetic differences [39,40]. Furthermore, interactions between *APOE* genotypes and mtDNA haplogroups may modulate the risk of developing AD [41,42]. Due to the strong linkage disequilibrium between *APOE* and *TOMM40*, interplay between *APOE* and mitochondria may converge at the level of mitochondrial function.

Currently, there is a lack of understanding regarding the molecular mechanisms by which mitochondria impact the development of AD or longevity. The involvement of the *APOE* locus in mitochondrial function may provide valuable insights into this matter. Therefore, our study aims to investigate changes in markers and genes related to mitochondrial structure and function using human cellular models induced with oxidative stress as well as PMBs from individuals with AD or normal controls. Our objective is to determine if there is a direct link between *APOE* locus genes and mitochondrial functions and to elucidate how this connection might explain the progression of AD and aging.

## 2. Results

### 2.1. Overview of the Study

To investigate the role of mitochondria in the cellular response to oxidative stress, we studied three models of human brain cells: an astrocyte-like cell line (U87), a microglia-like cell line (HMC3), and a neuron-like cell line (SH-SY5Y). To induce oxidative stress, we added hydrogen peroxide (H_2_O_2_) to the culture media and incubated the cells for 24 h (h) (referred to as “c1” for the first culture condition) before measuring mitochondrial biomarkers and gene expression. To assess the cells’ recovery from the H_2_O_2_ challenge, we replaced the treated culture media with fresh media and continued the culture for an additional 24 h (referred to as “c2”) or 48 h (referred to as “c3”) before performing various measurements. We compared the results of the treated cells to those of untreated cells. We further investigated changes in the selected cellular responses using PMB tissues to determine their relevance in AD. Table 1 provides demographic information for the PMB samples.

### 2.2. Oxidative Stress-Induced Cellular Responses in Mitochondrial Structure and Function

We first evaluated the impact of oxidative stress on mitochondrial structure and function by analyzing several biomarkers, including mitochondria membrane potential (MMP), mtDNA CN, and cell viability. MMP and mtDNA CN are quantitative indicators of mitochondria structural/functional integrity, while cell viability, assessed by the MTT assay, provides a measure of stress-induced toxicity. To account for differences in viable cell numbers, we normalized MMP and mtDNA CN assays across different cell lines and experimental conditions. Detailed procedures are described in the Materials and Methods section. At 24 h post-H_2_O_2_ treatment (c1), all three cell lines demonstrated reduced MMP (≈30–47%, Figure 1A), mtDNA CN (≈15–55%, Figure 1B), and cell viability (≈31–44%, Figure 1C). The glia-like cells (U87 and HMC3) exhibited a more significant reduction in mtDNA CN (>50%) than the neuron-like cell (SH-SY5Y), which had a milder reduction (≈15%). Similarly, SH-SY5Y showed the least reduction in MMP among the three cell lines. During the recovery phase, U87 cells restored both MMP and cell viability to their original state within 24 h (c2), while both recoveries were much slower in HMC3 cells and were not restored to the original state even after 48 h (c3). SH-SY5Y exhibited an MMP recovery pattern similar to those of U87 at two different recovery time points; however, its cell viability did not recover and further declined after 48 h (≈80%). All cell lines’ mtDNA CNs were restored to their original state after 48 h. Our results suggest that oxidative stress immediately decreases MMP, mtDNA CN, and cell viability in all three cell types, with a more acute impact on glia-like cells. The varying degrees of impairment and restoration indicate cell-type specificity in the oxidative stress response with astrocyte-like cells demonstrating robust mitochondrial recovery.

### 2.3. Oxidative Stress-Induced Alterations in Mitochondrial Structure and Function-Related Gene Expression

We next analyzed the expression profiles of genes associated with mitochondrial structure and functions, including antioxidant response genes (*REST*, *SOD1*, *SIRT1*), apoptotic pathway genes (*CASP3*, *CRYAB*), mitochondrial dynamics genes (*MFN1*, *FIS1*, *DNM1L*, *PINK1*), and a mtDNA maintenance gene (*TFAM*). We collected cells at 24 h post-H_2_O_2_ treatment (c1) and quantified mRNA by SYBR-based RT-qPCR, comparing RNA levels to untreated controls (Figure 2). While *TFAM* expression remained unchanged across all cells, SH-SY5Y cells showed an upregulation of three genes: *SOD1*, *SIRT1*, and *PINK1*. In contrast, all genes were upregulated in U87 with higher levels of antioxidant response genes (*REST*, *SOD1*) and mitochondrial dynamics genes (*MFN1*, *FIS1*). HMC3 cells showed slight upregulation or downregulation, except for significantly upregulated apoptotic genes (*CASP3*, *CRYAB*). We extended the recovery phase experiment on these genes, observing that *CASP3* and *CRYAB* in HMC3 cells returned to their original state after 24–48 h (Appendix A). The modest changes in expression levels of most genes suggest that these genes play a crucial role in mitochondrial homeostasis. Differential expression profiles suggest cell-type specificity, with astrocytes being the most sensitive to oxidative stress.

### 2.4. Oxidative Stress-Induced Alterations in APOE Locus Gene Expression

We extended our study to explore the potential role of the *APOE* locus genes in mitochondrial function. Specifically, we examined the mRNA expression profiles of four genes located within the *APOE* locus (*APOE*, *APOC1*, *NECTIN2*, and *TOMM40*) under the same experimental conditions of oxidative stress-induced cellular models. Our findings indicate that in general, all four genes were upregulated 24 h after H_2_O_2_ treatment in the three cell lines studied, except for *TOMM40* in SH-SY5Y, which showed an expression level similar to its untreated counterpart (Figure 3D). Notably, the upregulation of *APOC1* was most profound in HMC3, with an approximately 50% higher expression level than U87 and SH-SY5Y (Figure 3B). During the recovery phase, the expression levels of *APOC1* and *TOMM40* gradually returned to their original state, while *NECTIN2* remained upregulated, albeit with a small decline in U87 and SH-SY5Y after 48 h of recovery (Figure 3C). However, the expression of *APOE* was markedly different, remaining upregulated in HMC3 or continuously elevated in both U87 and SH-SY5Y as the recovery phase approached 48 h (Figure 3A). Importantly, unlike the mitochondria structure/function-related genes we previously tested, which exhibited random profiles of either upregulation or downregulation, all four *APOE* locus genes demonstrated the same trend of upregulation despite being associated with different biological pathways. This finding suggests a potential locus-specific coregulation of these *APOE* locus genes.

### 2.5. Mitochondrial Function Markers and Gene Expression in AD PMBs

As mitochondrial dysfunction is a key feature of AD, we sought to explore the gene expression profiles of AD PMBs to identify potential similarities with our oxidative stress-induced cellular models. Using frontal lobe tissues from AD and control subjects, we isolated total RNA and quantified mRNA levels of previously tested mitochondria-related genes. In a subset of PMB (14 AD and 10 Ctrl), we found either the upregulation or downregulation of these genes in AD brains (Appendix A), suggesting significant differences in mitochondria-related gene expression between AD and control brains. We also examined the expression profiles of *APOE* locus genes in the entire cohort of PMBs (73 AD and 27 Ctrl). Interestingly, all four *APOE* locus genes showed a unified pattern of upregulation in AD brains (Figure 4), suggesting a potential coregulation of these genes in response to physiological stimuli. We further quantified mtDNA CN in the entire PMB cohort without measuring other mitochondrial markers (MMP and MTT) that require live cells. We compared mtDNA CN measures with disease status as well as SNP alleles. These SNPs include rs429358 (*APOE*), rs11568822 (*APOC1*), and rs2075650 (*TOMM40*), which all have shown strong associations with both AD and longevity. Initially, we found no significant differences when stratified by disease status (Figure 5A) or SNP alleles (Appendix A). However, there was a statistically significant interaction term between disease status and the *APOE* SNP rs429358 (*p* = 0.027) on mtDNA CN, with the C allele (or ε4) being associated with greater mtDNA CN in AD brains (Table 2 and Figure 5B). These findings suggest that AD brains exhibit fundamental differences in mitochondrial function compared to control brains, which is possibly due to prolonged oxidative stress-induced mitochondrial dysfunction. Furthermore, our results suggest that the *APOE* locus may have a direct link with mitochondrial structure/function.

### 2.6. Three-Dimensional (3D) Genome Structure of the APOE Locus

The four *APOE* locus genes are involved in different biological pathways, making it challenging to understand how they are coregulated. However, advances in chromosome conformation capture technology have enabled high-resolution mapping of chromosome architecture, providing greater insight into supranucleosomal structures such as chromatin loops and topologically associating domains (TADs). By utilizing the UCSC Genome Browser’s (Hi-C and Micro-C Track Settings (ucsc.edu)) [43,44], we identified multiple interactions among the *APOE* locus genes (Appendix A). Additionally, several independent studies have also demonstrated chromosomal interactions within this *APOE* locus [45,46]. Our findings suggest that these four *APOE* locus genes are located within a single TAD, creating a 3D genomic environment that facilitates their coregulation. This coregulation likely leads to a combined biological effect that can be further influenced by various factors such as epigenetics, lifestyle, environment, and age.

## 3. Discussion

The strongest evidence connecting the *APOE* ε4 allele with AD risk has been shown through the studies of human genetics and epidemiology. It is most widely accepted that *APOE* ε4 increases AD risk by promoting Aβ aggregation and impeding its clearance, while other factors such as neuroinflammation and blood–brain barrier integrity may also be involved [47]. However, the specific mechanisms by which inheriting the *APOE* ε4 allele translates into AD risk and how this risk is manifested are still not fully understood. Although several *APOEs* targeting therapeutic strategies have been proposed [48], none of the *APOE-*related AD intervention strategies have been currently proven to be useful. Our study introduces a new concept, which involves multiple genes in the *APOE* locus, to explain how this genetic context could impact AD and aging. Such a concept could provide a clearer direction to design intervention strategies in the future.

Genetic studies have shown that the *APOE* locus contains several genes that are strongly linked to either an increased risk of AD or longevity. There are two hypotheses that could explain this observation. The first hypothesis proposes that *APOE* is the only effector, and other genes’ signals are surrogates due to high linkage disequilibrium in the region. In contrast, the second hypothesis suggests that multiple genes in this locus have combined effects that contribute to an outcome of either AD or longevity. Although the former single-effector hypothesis proposes that *APOE* alone is sufficient to drive the biological consequences, given that the primary biological function of the APOE protein is in lipid metabolism, it cannot easily explain the diverse physiologies of AD or longevity. On the other hand, the latter multi-effector hypothesis provides a more compelling explanation, adding *APOC1* (a component of innate immunity) and *TOMM40* (a component of mitochondrial function). Despite the single-effector hypothesis dominating AD research for the last 30 years, the multi-effector hypothesis has become increasingly appealing due to advances in 3D genome technologies, which facilitate comprehension of the multiple genes’ coregulation within the same locus. Since the *APOE* locus is associated with AD and aging, and mitochondrial dysfunction is a common feature between AD and unsuccessful aging, we conducted this study using cellular models and PMBs to investigate the connection between the *APOE* locus and oxidative stress-related mitochondrial function. Our findings support the multi-effector hypothesis in the *APOE* locus.

Oxidative stress can induce alteration of mitochondrial markers and gene expression to alleviate cellular defects and work as a compensatory mechanism for the stress. Oxidative stress can also lead to mitochondrial dysfunction and cell death through activation of the apoptotic pathway [49,50]. To investigate oxidative stress-induced changes in mitochondrial structure/function-related markers, we employed cellular models. Studies of functional genomics have shown that gene regulation varies substantially across tissues and cell types [51]. Within this context, the pathophysiology of AD and aging has been detected in three brain-associated cell types including astrocytes, microglia, and neurons. Hence, we selected these three cell types-like cell lines in our study. Furthermore, among the brain cells, *APOE* is mainly expressed in glia cells and expressed in neurons only under stress conditions. Due to the limited availability of human brain-related primary cell lines, the immortalized cell lines (i.e., U87, HMC3, SH-SY5Y) that express characteristics of human brain cells provide an alternative. These cellular models have been extensively utilized to study molecular mechanisms underlying the pathophysiology of aging and aging-related neurodegenerative diseases [52]. Our data indicate that H_2_O_2_-induced oxidative stress immediately reduced mitochondrial membrane potential, mtDNA CN, and cell viability in all three cellular models tested. During the recovery phase, all reduced properties except cell viability in SH-SY5Y returned to their original state. This may be due to the culture characteristics of the SH-SY5Y cell line, which includes both adherent and floating cells [53], with the latter being discarded during media changes in the recovery phase. Additionally, the mixed cell types of SH-SY5Y, including both proliferative epithelial and differentiated neuron-like cells, may also account for the decreased number of viable cells during recovery [54,55]. Among glial cells, U87 cells demonstrated the most robust recovery response, consistent with the known roles of astrocytes in various structural, metabolic, and homeostatic functions in the central nervous system (CNS) [56]. In contrast, HMC3 cells showed a slower recovery rate, which may reflect the innate immunity role of microglia in transforming themselves to counter stress-induced inflammation in the CNS. Our findings suggest that the oxidative stress response exhibits robust cell-type specificity, as evidenced by varying degrees of reduction and restoration of mitochondrial function markers.

We also investigated the effects of oxidative stress on mitochondria-related gene expression. We focused on a subset of genes commonly associated with mitochondrial function and found that U87 consistently upregulated all genes during the initial oxidative stress, while HMC3 and SH-SY5Y showed mixed responses, indicating cell-specific responses to oxidative stress. In U87, the fusion/fission pathway was quickly activated, as seen by the significant upregulation of genes responsible for mitochondrial fusion and fission, *MFN1* and *FIS1*. This suggests that astrocytes can quickly collaborate to address stress and maintain mitochondrial integrity. HMC3 upregulated apoptotic pathway genes, *CASP3* and *CRYAB*, to counter stress. *CRYAB* has known to be a sensitive marker for oxidative stress [57,58]; therefore, apoptosis might be one of the primary responses when microglia encountered oxidative stress in the brain. On the other hand, SH-SY5Y downregulated *CASP3*, mitigating apoptotic activation that may be beneficial for the survival of postmitotic neuron cells. SH-SY5Y also upregulated mitophagy-associated gene *PINK1*, which is a sensor for damaged mitochondria and initiates mitophagy. This suggests that postmitotic neurons prefer mitophagy over apoptosis to maintain mitochondrial integrity. Our results show that oxidative stress causes immediate effects on mitochondrial structure and function, leading to acute changes in gene expression, which returns to baseline during the recovery phase to maintain homeostasis. Cell-specific responses were also evident in this process.

Our study of the four *APOE* locus genes using cellular models yielded some surprising results. After an initial challenge of oxidative stress, we consistently observed the upregulation of all four genes in all cell lines tested, which stands in stark contrast to previous experiments where we observed both upregulations and downregulations of mitochondrial function-related genes. During the recovery phase, the RNA levels of all upregulated *APOE* locus genes gradually returned to their untreated state except for *APOE*. Its RNA level continued to rise even after the oxidative stress was removed, reinforcing the idea that the APOE protein plays a crucial role in providing lipids that alleviate stress-induced damage as well as influencing mitochondrial biogenesis/dynamics and immunity [59,60]. Notably, we found that the RNA level of *APOE* increased the most in HMC3 cells, and the increase persisted during the recovery phase. This could be explained by *APOE‘*s link in immunity. APOE is a ligand for binding to TREM2, which activates signaling pathways in microglia for cell survival, proliferation, phagocytosis, and immune responses [61,62]. Since mitochondria are a crucial component of the immune system [63], there is a natural link between mitochondria and *APOC1*. The upregulation of *APOC1* is associated with macrophage activation and possibly microglia [64], which may explain the observed upregulation of *APOC1* in HMC3 cells under oxidative stress. Furthermore, we found that *NECTIN2* remained upregulated in both U87 and SH-SY5Y cells even after a 48 h recovery phase. *NECTIN2* encodes a single pass type I membrane glycoprotein that serves as an entry point for the herpes simplex virus and a signaling modifier in immune cell responses, indicating a potential association with mitochondria-related immune responses [65,66]. In contrast to the other three *APOE* locus genes, the upregulation of *TOMM40* was observed to a much smaller extent in all three cell lines, suggesting that *TOMM40* may function as a housekeeping gene that is stably expressed and not subject to much fluctuation under different stimuli [67]. The TOM40 protein forms the main entry gate of the outer mitochondrial membrane (TOM) complex, which is responsible for the transport of all nucleus-encoded mitochondrial proteins. Therefore, genetic variants of *TOMM40* that impair transport efficiency over the long term could compromise mitochondrial function. Indeed, *TOMM40* variants have been associated with neuroinflammation and mitochondrial dysfunction, which can increase the risk of developing AD [68]. Moreover, the regulatory effects of *TOMM40* variants could also affect the expression of both *TOMM40* and *APOE* [69], indicating a potential link between *TOMM40* and *APOE*. Taken together, our findings suggest that *TOMM40* may play a critical role in maintaining mitochondrial structure and function in response to oxidative stress, whereas *APOE* acts as a secondary responder with sustained upregulation to repair cellular damages. *APOC1* and *NECTIN2* may also function as secondary responders, activating the immune system to clear damaged cells or misfolded proteins in response to oxidative stress. Interestingly, all four *APOE* locus genes showed similar patterns of upregulation in all three cell lines in response to acute oxidative stress, suggesting the possibility of coregulation among these genes. This result also sheds new light on the complex interplay between *APOE* locus genes in modulating cellular responses to oxidative stress.

Our study on PMBs reveals a mixed pattern of either upregulation or downregulation of mitochondria-related gene expression in AD brains when compared to control brains. In contrast, we observed a uniform upregulation of all four *APOE* locus genes in the AD brains, suggesting a potential coregulation of these genes under various physiological conditions. Moreover, our findings on mtDNA CN suggest that the progression of AD disease could affect mtDNA CN in the brain, and this effect could be modulated by the *APOE* SNP. MtDNA CN has been associated with various human diseases [70] and linked with genetic variants of the *APOE* locus [71]. Our results strengthen the rationale that multiple genes in the *APOE* locus play a direct role in mitochondrial function. As mitochondria play a crucial role in both AD and aging, their integrity and function may serve as a common intersection that mediates biological pathways and modifies physiology between the two. Damage to brain mitochondria can result in energy production deficiencies, oxidative stress, and the generation of mitochondria-derived damage-associated molecular patterns that trigger inflammation and neuronal damage [72,73]. Considering that mitochondrial dysfunction is a hallmark of neurodegeneration and aging, analyzing differential expression profiles and mtDNA CN between AD and control PMBs could provide valuable insights into the long-term and cumulative changes induced by oxidative stress.

Recent advances in chromosome conformation capture technology [74], including its high-throughput version (Hi-C) [75], have facilitated the high-resolution mapping of chromosomal interactions and loops that serve as the building blocks of 3D genomes. The genome is hierarchically folded and segmented into TADs [76,77]. Genes located within the same TAD region possess regulatory elements that interact with each other to form a complex gene regulatory network [78,79,80]. Epigenetics, lifestyle, environment, and age can modify such regulation, leading to changes in cellular states and physiology. As such, TADs can be considered the fundamental functional unit of the genome. Recently, research in AD has applied this concept [81,82]. Based on evidence from chromosome conformation capture [44,45,46], it is likely that multiple genes within the *APOE* locus are coregulated within the same TAD. Therefore, TADs are well-suited to study the mechanisms of coregulation of multiple genes within the *APOE* locus. Genetic variations within this region can have combined and long-term effects that modify aging-related decline in cholesterol transport (*APOE*), immunity (*APOC1*, *NECTIN2*), and mitochondrial function (*TOMM40*). With the presence of suboptimal genetic variants, this locus’ detrimental effect may lead to cholesterol and myelin deficiencies in the CNS, impairing neuronal recovery after various damages, including infections. They may also result in the accumulation of misfolded proteins due to inadequate energy production and compromised innate immunity. Thus, genetic haplotypes variants of the *APOE* locus may either accelerate or mitigate the accumulation of brain damage with age, leading to the development of AD or successful aging.

While our study sheds light on the acute phase of the oxidative stress response and recovery, there are several limitations that warrant acknowledgement. Firstly, the cellular models we employed only captured a partial aspect of mitochondrial dynamics, and our experiments did not provide a comprehensive perspective of mitochondrial biology reflecting disease and aging due to the limited data on timing and gene pathways. Secondly, the PMB samples we collected were cross-sectional, providing only a snapshot of mitochondrial changes in the later stages of the lifespan. Therefore, our findings may not fully represent the entire process of mitochondrial dysfunction during aging and disease. Addressing these limitations would require further studies that utilize longitudinal data with larger sample sizes to provide a more comprehensive understanding of mitochondrial biology in the context of aging and disease.

## 4. Materials and Methods

### 4.1. Human PMB and Cell Lines

Human biospecimen were obtained from the University of Washington (UW) Alzheimer’s Disease Research Center (ADRC) after approval by the institutional review board of the Veterans Affairs Puget Sound Health Care System (MIRB# 00331). All human PMB used in this study have been obtained from the UWADRC Brain Bank. The UWADRC has IRB approval to collect tissues through HSRC# 03-7902-D, entitled, “Autopsy and Neuropathology Core D: Alzheimer’s Disease Research Center” (Thomas Grabowski, PI). In addition, this study involves autopsy material from deceased individuals, which does not meet the regulatory definition of “human subject research”. AD patient diagnosis was confirmed postmortem by neuropathological analysis. Clinically normal subjects were volunteers over 65 years of age, never diagnosed with AD, and lacked AD neuropathology at autopsy. Demographics of the PMB samples are listed in Table 1. Postmortem frontal lobe tissues were obtained from the middle frontal gyrus tissues that were rapidly frozen at autopsy (<10 h after death) and stored at –80 °C until use. Glioblastoma U87 MG cells (ATCC, Manassas, VA, USA) grew in 89% Dulbecco’s modified Eagle’s medium (DMEM) (Gibco, ThermoFisher, Waltham, MA, USA). Microglia cells used in this study were human microglial clone 3 cell line, HMC3 (ATCC, Manassas, VA, USA), which was established through the SV40-dependent immortalization of human embryonic microglial cells [83] extensively distributed under the name of CHME3 and authenticated by ATCC [52]. HMC3 cells were grown in 89% Eagle’s Minimum Essential Medium (EMEM) (ATCC, Manassas, VA, USA); neuroblastoma SH-SY5Y cells (ATCC, Manassas, VA, USA) were grown in 89% DMEM with F12. All these media were supplemented with 1% penicillin/streptomycin and cells cultured at 37 °C in a 5% CO_2_ atmosphere.

### 4.2. Hydrogen Peroxide Treatment

Twenty-four hours prior to treatment, three cell lines (U87, HMC3 and SH-SY5Y) were seeded at a density of 70–80%. For the assays of mtDNA CN and gene expression, cells were seeded on a 6-well plate, while a 96-well plate was used for mitochondrial membrane potential assay and cell viability assay. The optimal concentration of hydrogen peroxide, H_2_O_2_ (Sigma, Burlington, MA, USA), for each cell line was determined when over 70–80% cells remained healthy during 24 h treatment. Then, 600 µM (U87), 400 µM (HMC3), or 100 µM (SH-SY5Y) H_2_O_2_ was added to the cell culture and incubated for 24 h at 37 °C, 5% CO_2_ (designated “c1” for the first culture condition). In addition, cells were assayed during the recovery phase of H_2_O_2_-induced oxidative stress. To do this, cells were treated with H_2_O_2_ for 24 h, after which cell media was replenished with fresh growth media and cells were incubated for 24 h (designated “c2”) or 48 h (designated “c3”). For each condition, we set up the control without treatment and compared it with treated conditions. The treated and untreated cells were collected and subjected to genomic DNA and total RNA isolation. Three to four independent experiments were performed.

### 4.3. DNA/RNA Extraction

Genomic DNA and RNA were isolated from cultured cell lines and frozen PMB tissues from the frontal cortex. Genomic DNA was extracted using the QIAmp DNA Blood Mini Kit (Qiagen, Hilden, Germany) and RNA was extracted using an AllPrep DNA/RNA Mini Kit (Qiagen, Hilden, Germany) according to the manufacturer’s protocols. Nucleic acid concentrations were measured by NanoPhotometer (Implen, Westlake Village, CA, USA), and samples were stored at –20 °C prior to use.

### 4.4. Mitochondrial Membrane Potential (MMP) Assay

The MMP of the human cell lines was measured using a MitoProbe JC-1 assay kit (ThermoFisher, Waltham, MA, USA). The cationic dye, JC-1 (5′,6,6′-tetrachloro-1,1′,3,3′-tetraethylbenzimidazolylcarbocyanine iodide), exhibits membrane potential-dependent accumulation in mitochondria, which is indicated by a fluorescence emission shift from monomeric green (529 nm) to JC-1 aggregates red (590 nm). Consequently, the MMP change in response to cellular stimuli is represented by the ratio of red to green fluorescence intensity. This ratio of red dye aggregated in mitochondria to monomeric green dye was used to account for variations in the total number of viable cells in experimental conditions. The membrane potential disrupter, CCCP (carbonyl cyanide 3-cholorophenylhydrazone), was included in all assays as a control. First, 24 h prior to H_2_O_2_ treatment, cells were seeded at a density of 70–80% on a 96-well black plate with a clear bottom. The seeded cells were treated with H_2_O_2_ for 24 h and then proceeded to MMP assay. After washing cells on the plate with growth media, 2 µM JC-1 was added and incubated at 37 °C, 5% CO_2_ for 30 min. The reaction plate was washed with PBS, and fluorescence was measured in 488 nm excitation and green (529 nm) or red (590 nm) emission using a SpectraMax M2 plate reader (Molecular Devices, San Jose, CA, USA). All procedures were performed according to the manufacturers’ protocols. The MMP fold change (FC) of H_2_O_2_-treated to untreated was computed as FC (treated) = MMP (treated)/MMP (untreated). Five to six independent experiments were performed.

### 4.5. Mitochondrial DNA Copy Number (mtDNA CN) Assay

MtDNA CN was quantitated by SYBR-based quantitative PCR (qPCR). Mitochondria-encoded NADH dehydrogenase 1 (*MT-ND1*) copy numbers were quantitated and normalized by the copy numbers of a single copy reference gene, nuclear receptor coactivator 3 (*NCOA3*). Reactions for *MT-ND1* and *NCOA3* were run separately in a 384-well optical plate in triplicate using QuantStudio 5 (ThermoFisher, Waltham, MA, USA). Since there are so many copies of mtDNA compared to the single-copy reference gene, optimal amounts of genomic DNA need to be determined so that the cycle threshold (Ct) is in the linear range of the amplification curve. First, 0.1 ng of genomic DNA for *MT-ND1* and 3.3 ng for *NCOA3* were used in qPCR assays. As the constant amount of input DNA was used for mtDNA CN measurement, variations in the total number of viable cells were taken into account when analyzed in experimental conditions. The 10 µL reaction included respective amounts of DNA for *MT-ND1* or *NCOA3*, 5 µL of Power SYBR GREEN PCR Master Mix (Applied Biosystems, ThermoFisher, Waltham, MA, USA), and 0.5 µM of each forward and reverse primer. The thermal cycling profile consisted of 2 min at 50 °C, 10 min at 95 °C, and then 40 cycles of 15 s at 95 °C, 60 s at 56 °C, and 60 s at 72 °C. The ΔCt method was used to control for the quantity of input DNA by the single-copy reference gene *NCOA3*. The normalized ΔCt = mean of *NCOA3* Ct triplicate − Ct (*MT-ND1*). In this setting, a larger ΔCt value indicates a higher mtDNA CN. In addition, the fold change of H_2_O_2_-treated to untreated was computed as FC (treated) = 2^−ΔΔCt^, where ΔΔCt = ΔCt (treated) − ΔCt (untreated).

### 4.6. Cell Viability Assay

MTT (3-(4,5-dimethylthiazol-2-yl)-2,5-diphenyltetrazolium bromide) (Abcam, Waltham, MA, USA) assay was performed for measuring cell viability, cell proliferation and cytotoxicity. This assay is based on the conversion of a water-soluble MTT compound to an insoluble formazan by metabolically active cells, resulting in color formation at OD590 nm. Dead cells lose this ability and show no signal. The measured absorbance at OD590 nm is proportional to the number of viable cells. FC (treated) = OD590 (treated)/OD590 (untreated).

### 4.7. Gene Expression by Reverse Transcriptase (RT) Reaction and Quantitative PCR (qPCR) Assay

Total RNA (100 ng) was used for each 20 μL RT reaction, and cDNA synthesis was performed using the PrimeScript RT Reagent Kit (Takara Bio USA, Mountain View, CA, USA). The resulting cDNA was diluted 20 times for qPCR, and expression levels were measured using TaqMan assays in a QuantStudio 5 (ThermoFisher, Waltham, MA, USA). Each 10 μL qPCR reaction contained a fixed RNA input (5 ng), 0.5 μL of the 20X TaqMan assay, and 5 μL of 2X TaqMan Universal PCR Reaction Mix (Applied Biosystems, Thermo Fisher). The thermal cycling program consisted of 2 min at 50 °C, 10 min at 95 °C, and then 40 cycles of 15 s at 95 °C and 1 min at 60 °C. For mitochondrial structure and function-related gene expression, SYBR PCR reactions were run on a QuantStudio 5 (ThermoFisher, Waltham, MA, USA) with the same thermal cycling program as used for the TaqMan assays. Triplicate qPCR assays were performed for all gene expression experiments. To control for the quantity of input RNA, we quantified *ACTB* mRNA as an internal control for each sample and obtained a normalized ΔCt value: mean of the *ACTB* Ct triplicate—Ct (target). In this setting, ΔCt values were near or below zero, as *ACTB* mRNA levels were higher than those of target genes; a larger ΔCt value indicates a higher RNA level. Additionally, the fold change of H_2_O_2_-treated to untreated was computed as FC (treated) = 2^−ΔΔCt^, where ΔΔCt = ΔCt (treated) − ΔCt (untreated). Information on primers, probes and TaqMan assays is listed in Appendix A.

### 4.8. TOMM40 Allelic Gene Expression

For the measurement of *TOMM40* RNA levels in the PMB sample, the quantification of allelic gene expression was performed by pyrosequencing combined with duplex digital PCR. Previously, we have shown that a commercial TaqMan assay was not suitable for the accurate measurement of *TOMM40* RNA, which was likely due to the presence of *TOMM40* pseudogenes RNA [11]. For the *TOMM40* allelic gene expression assay, first, a DNA segment of *TOMM40* spanning exon 3 and exon 4 was PCR-amplified using a biotinylated forward primer and a reverse primer in which the strand to serve as the pyrosequencing template is biotinylated. After denaturation, the biotinylated single-stranded PCR amplicon was subjected to hybridize with a sequencing primer. The pyrogram signal peak for the *TOMM40*-specific allele distinct from the *TOMM40* pseudogene allele was represented as percentage. Pyrosequencing was carried out on a PyroMark Q24 system (Qiagen, Hilden, Germany), and data were analyzed using PyroMark Q24 software, version 2.0.6 (Qiagen, Hilden, Germany). Then, the copy number of the *TOMM40* total RNA including the *TOMM40* pseudogene RNA was quantified by a duplex digital PCR. Two pairs of primer and probe mix corresponding to *TOMM40* and *ACTB* were run together in a reaction using QIAcuity dPCR (Qiagen, Hilden, Germany). All procedures were performed according to the manufacturers’ protocols. Using *ACTB* as an endogenous control, the *TOMM40* RNA copy number was normalized by *ACTB* RNA copy number. The final computed *TOMM40* RNA copy number: the normalized *TOMM40* RNA copy number × the percentage of the *TOMM40* allele. Information on primers, probes and TaqMan assays is listed in Appendix A.

### 4.9. Genotyping of APOE, TOMM40 and APOC1

Genomic DNA isolated from frozen PMB samples was used for genotyping. The ε2/ε3/ε4 alleles of *APOE* single nucleotide polymorphism (SNP)*,* rs429358, were genotyped using two TaqMan allele discrimination assays (C_3084793_20 and C_904973_10, ThermoFisher, Waltham, MA, USA). The *TOMM40* SNP, rs2075650, was genotyped using TaqMan allele discrimination assays (C_3084828_20 and C_31478296_10, ThermoFisher, Waltham, MA, USA). The *APOC1* SNP, rs11568822 (In/del), was genotyped by restriction fragment length polymorphism (RFLP). A small fragment enclosing the *APOC1* SNP region was amplified by a standard Hot Start PCR with a primer pair: Ch19_50109453F (5′-ATTCCCCGAACGAATAAACC 3′) and Ch19_50109512R (5′ AGCCGCAGACAAAATTCCT-3′) (Integrated DNA Technologies, Coralville, IA, USA). The PCR fragment was digested with the enzyme HpaI (cut site GTT^AAC) and analyzed for the fragment size difference between insertion (CGTT) and deletion using QIAxcel with a DNA High-Resolution Cartridge (Qiagen, Hilden, Germany).

### 4.10. Statistical Analysis and Box Plot

The figures were created using the software R-Program Version R-4.2.2 for Windows. The package ggpubr (CRAN—Package ggpubr (r-project.org)) was used to generate the box plots. The package rstatix (CRAN—Package rstatix (r-project.org)) was used to perform an independent samples *t*-test. Statistical analyses including linear mixed effects model and pairwise comparison were performed on IBM SPSS Statistics 19 for Windows, Version 19.0 (IBM Corp., Armonk, NY, USA).

## 5. Conclusions

The *APOE* locus harbors several genes that could be coregulated. Among these genes, *APOE* participates in lipid transport and the clearance of neuropathological aggregates, *APOC1* and *NECTIN2* regulate innate immunity, and *TOMM40* facilitates mitochondrial protein transport. These genes are most likely located within the same TAD of the 3D chromosome conformation, enabling an efficient coregulatory response to environmental cues and insults, particularly when the locus genes are involved in multiple pathways. The collective impact of these genes could modulate mitochondrial biology, ultimately leading to either the development of AD or a path toward healthy aging.

## Figures and Tables

**Figure 1 ijms-24-10440-f001:**
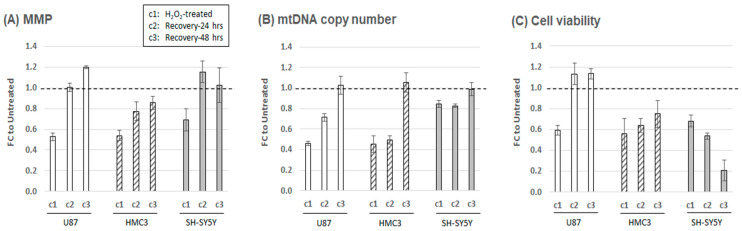
Oxidative stress-induced cellular responses in mitochondrial structure and function. Assays for mitochondrial membrane potential (MMP) (**A**), mitochondrial DNA (mtDNA) copy number (**B**), and cell viability (**C**) were performed on three cellular models. The fold changes (FC) of experimental conditions are compared to their untreated counterparts (set at 1.0, dashed line) and plotted as the average and standard deviation of five to six independent experiments. The first culture condition, c1, represents cells treated with H_2_O_2_ for 24 h; c2, cells in H_2_O_2_ recovery phase after replenishing culture with fresh media and continuing to culture for additional 24 h; c3, cells in H_2_O_2_ recovery phase for culturing additional 48 h.

**Figure 2 ijms-24-10440-f002:**
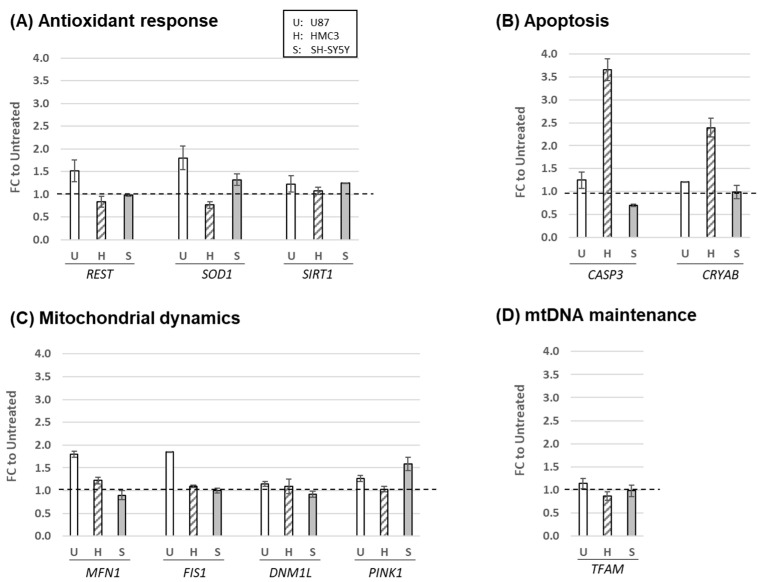
Oxidative stress-induced alterations in mitochondrial structure and function-related gene expression. RNA levels of three cellular models treated with H_2_O_2_ for 24 h were quantified by RT-qPCR (SYBR-based). (**A**) Antioxidant response genes, (**B**) apoptosis genes, (**C**) mitochondrial dynamics genes, and (**D**) a mtDNA maintenance gene. The fold changes (FC) of H_2_O_2_-treated cells to the untreated counterpart (set at 1.0, dashed line) are plotted with average and standard deviation from three to four independent experiments.

**Figure 3 ijms-24-10440-f003:**
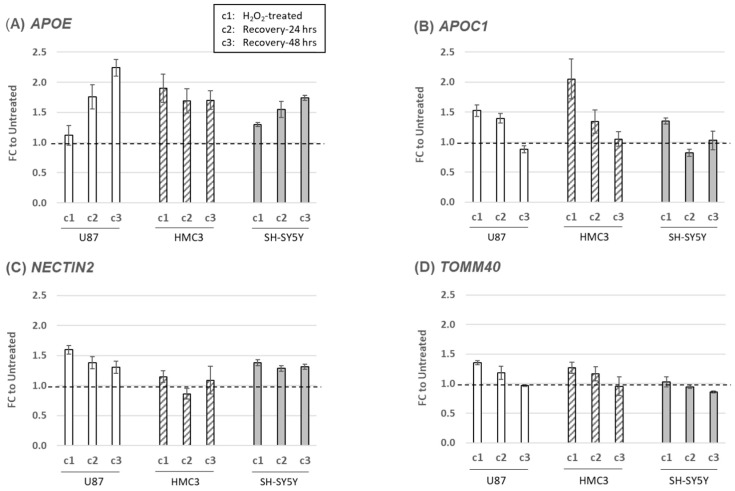
Alterations of *APOE* locus gene expression in response to oxidative stress. RNA levels of four genes in the *APOE* locus [*APOE* (**A**), *APOC1* (**B**), *NECTIN2* (**C**), and *TOMM40* (**D**)] in experimental conditions were measured by RT-qPCR using TaqMan assays. Fold changes (FC) to their untreated counterparts (set as baseline of 1.0) are plotted as average and standard deviation of three to four independent experiments. The first culture condition, c1, represents cells treated with H_2_O_2_ for 24 h; c2, cells in H_2_O_2_ recovery phase after replenishing culture with fresh media and continuing to culture for additional 24 h; c3, cells in H_2_O_2_ recovery phase for culturing an additional 48 h.

**Figure 4 ijms-24-10440-f004:**
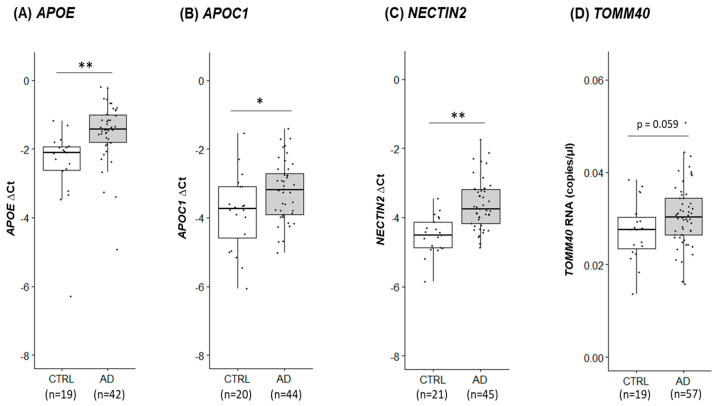
Variations of *APOE* locus gene expression in human PMB tissues. RNA levels of four genes in the *APOE* locus were quantified by RT-qPCR (TaqMan assay) in PMB samples from AD and control (CTRL): *APOE* (**A**), *APOC1* (**B**), and *NECTIN2* (**C**). For *TOMM40* (**D**), digital PCR was performed and shown as RNA copies/µL. The ∆Ct method was used: a larger ΔCt value indicates a higher RNA level. An independent samples *t*-test was used to compare AD and control. *, *p* < 0.05; **, *p* < 0.005. Numbers in parentheses denote PMB sample sizes.

**Figure 5 ijms-24-10440-f005:**
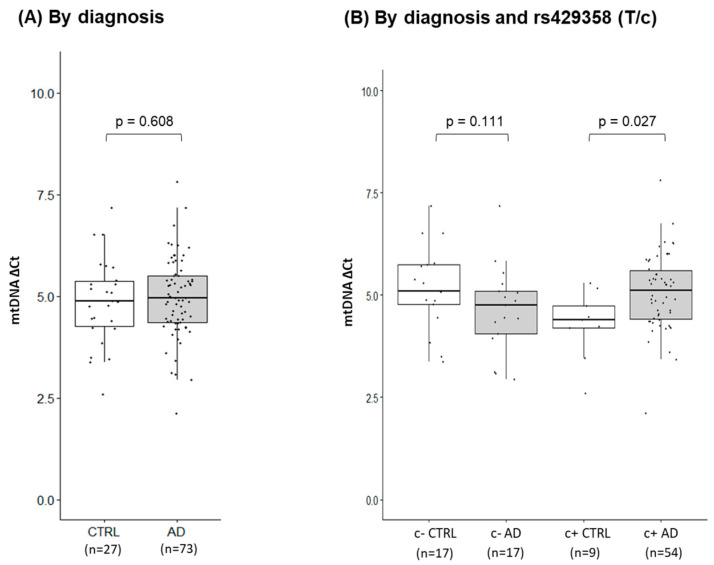
Human PMB mitochondrial DNA (mtDNA) copy numbers stratified by diagnosis and *APOE* genotype. Using PMB AD and control (CTRL) samples, mtDNA copy numbers were quantified by RT-qPCR (SYBR-based). mtDNA copy numbers are stratified by diagnosis (**A**) and alleles of *APOE* SNP rs429358 (**B**). The normalized ΔCt shows that a larger ∆Ct value indicates a greater number of mtDNA. *p*-values from a linear mixed effects model and pairwise comparison with the covariate, *APOE* ε4 status. c- indicates subjects with no ε4 alleles; c+ indicates subjects with at least one ε4 allele.

**Table 1 ijms-24-10440-t001:** Demographics of the PMB study samples.

Subjects	AD	Control
Sample number_n	73	27
Gender-Female_n (%)	41 (56.2)	15 (55.6)
APOE e4_n (%)	55 (75.3)	9 (33.3)
Age at death_mean (SD)	85.88 (7.3)	88.44 (7.5)
Age at onset_mean (SD)	76.34 (9.6)	N/A
Disease Duration_mean (SD)	9.55 (4.8)	N/A
Postmortem interval_mean hour (SD)	5.06 (1.8)	5.04 (2.4)
CERAD Score		
Absent	0	9
Sparse	0	9
Moderate	11	7
Frequent	62	2
BRAAK Stage		
I	0	5
II	0	12
III	0	10
IV	0	0
V	20	0
VI	53	0

**Table 2 ijms-24-10440-t002:** Effects of SNP rs429358 in PMB mtDNA CN by a linear mixed effects model analysis.

Without Covariates		
	Mean (SD)	
	CTRL	4.730 (0.594)
	AD	4.855 (0.576)
	Mean Difference (SE)	0.124 (0.241)
	*p*-value	0.608
	95% CI	[−0.354, 0.603]
With Covariates		
	Mean (SD)	
	c- CTRL	5.178 (0.607)
	c- AD	4.643 (0.607)
	c+ CTRL	4.283 (0.646)
	c+ AD	5.067 (0.575)
	Mean Difference (SE)	
	c- (CTRL vs. AD)	0.535 (0.333)
	*p*-value	0.111
	95% CI	[−1.195, 0.125]
	Mean Difference (SE)	
	c+ (CTRL vs. AD)	0.783 (0.349)
	*p*-value	0.027
	95% CI	[0.090, 1.476]

## Data Availability

The data presented in this study are available on request from the corresponding authors.

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
