# Peer review of "APOE Locus-Associated Mitochondrial Function and Its Implication in Alzheimer’s Disease and Aging"

_ijms, 2023, doi:10.3390/ijms241310440_

Round 1

Reviewer 1 Report

I have completed my review of manuscript ijms-2397016.

Alzheimer's disease (AD) is progressive neurodegenerative disorder that cause dementia and eventually death. There is currently no efficient treatment available to slow or stop the progression of AD. The causes of AD are uncertain, and there is no treatment. The APOE locus has attracted considerable clinical attention due to its connection with Alzheimer's disease (AD) and longevity. This genetic association is observed in various genes within the APOE locus. This study aimed to examine alterations in markers related to mitochondrial structure and function using human cellular models induced by oxidative stress, as well as postmortem brains (PMBs) obtained from individuals with Alzheimer's disease (AD) and normal controls. The findings unveiled a diverse range of gene expression changes, showing both upregulation and downregulation, in response to oxidative stress. This study highlights the crucial role of the APOE locus in modulating mitochondrial function and offers valuable insights into the underlying mechanisms of Alzheimer's disease (AD) and aging. It underscores the significance of this locus in clinical research, emphasizing its importance for further understanding and investigating AD and aging-related processes.

In my opinion, the review is important and timely to address some important factors. Overall, I am positive about this manuscript. Before making a positive decision, I have some concerns and comments about the present form of the manuscript that must be addressed first.

Comments for authors

Comment 1: AD can be caused  by a variety of factors. The introduction might need some addition of recent information. Recently, the microwave exposure was also thought to be responsible for AD. I encourage authors to add some background on this topic for readers in the introduction section. The suggested article may assist authors in expanding their background knowledge and understanding the mechanisms by which the EM field interacts with and affects biological systems for various effects.

Article: Microwave Radiation and the Brain: Mechanisms, Current Status, and Future Prospects. International Journal of Molecular Sciences vol. 23 (2022). [https://doi.org/10.3390/ijms23169288].

Comment 2: Authors indicate the unit “hours” and many timesa as “hr” in their explanation. Be specific throught the manuscript.

Comment 3: How do APOE gene variants, particularly APOE ε4, influence the risk of developing AD?

Comment 4: What specific role does the APOE locus play in modulating mitochondrial function?

Comment 5: What is the relationship between APOE gene variants and neuroinflammation in AD and aging?

Comment 5: re there any potential interventions or strategies that could target the effects of the APOE locus on AD and aging-related processes?

Comment 6: The paper contains errors and typos. I encourage authors to reread carefully and fix any grammatical errors.

The paper contains errors and typos. I encourage authors to reread carefully and fix any grammatical errors.

Reviewer 2 Report

The manuscript is focused on the APOE locus and its implication in AD and longevity. Authors examed changes in mitochondrial structure/function-related markers using oxidative stress-induced human 16 cellular models and postmortem brains (PMBs) from individuals with AD and normal controls. They found expressional alterations, either up- or down-regulated, in these genes in 18 response to oxidative stress. Also they observed an upregulation of multiple APOE 19 locus genes in all cellular models and AD PMBs. Additionally, the effects of AD status on mitochon- 20 drial DNA copy number (mtDNA CN) varied depending on APOE genotype. Those data are clear and enough to publish to IJMS.

Reviewer 3 Report

The manuscript entitled “Mitochondrial Function-Associated APOE Locus and its Implication in Alzheimer’s Disease and Aging” by Lee and coworkers is a well-written and well-executed study that resulted several novel findings.

I have read the manuscript with great interest and appreciated the effort made by the authors. The manuscript is excellently written. The Introduction is informative with plenty of bibliographical references; in fact, more than half of the references are listed here. Despite this, some further references justifying the use of the specific cell lines should be included. The Results are clearly presented. Perhaps some of the legends/numbering on the graphs could be made larger for better viewing. The Discussion is thorough and relevant to the results obtained. The methodology of the research is sound, the Materials and methods section gives sufficient detail on the biomaterials and protocols used. However, there are a few minor problems that the authors should reflect upon in the revised manuscript.

Minor problems

1) Lines 99-100: The authors state that the cell line SH-SY5Y is a “neuron-like cell line”. The SH-SY5Y cell line is a thrice cloned subline of the neuroblastoma cell line SK-N-SH (ATCC HTB-11), which was established in 1970 from a metastatic bone tumor from a 4-year-old cancer patient (see  https://www.atcc.org/products/crl-2266 for details). Similarly, the cell line U87 MG is of glioblastoma origin. Further, the origin of the cell line HMC3 is also interesting (ATCC says: “HMC3 is a microglial cell that was isolated from the brain of a patient. This cell line was deposited by KH Krause (University of Geneva, Switzerland)”. Dello Russo et al. (2018) states that “HMC3, was established in 1995, through SV40-dependent immortalization of human embryonic microglial cells”. Please discuss the reference below in relation to choosing this cell line to the aims of this study.

Dello Russo C, Cappoli N, Coletta I, Mezzogori D, Paciello F, Pozzoli G, Navarra P, Battaglia A. The human microglial HMC3 cell line: where do we stand? A systematic literature review. J Neuroinflammation. 2018 Sep 10;15(1):259. doi: 10.1186/s12974-018-1288-0. PMID: 30200996; PMCID: PMC6131758.

Why the authors think that these immortalized or tumor cell lines are good representations or models of either the normal physiology or an Alzheimer’s pathophysiology? I think this should be explained in the Introduction section or discussed later in detail (with references).

2) Lines 423-425: I believe that even if the collection of human samples was done by others, the original institutional (ethics) approval (stating the original purpose) should also be listed here as human sample collection, analysis, data handling, etc. cannot be published without a documented approval by an authority. The reference to the original study that first used these biomaterials should be given in the Materials and methods section.

3) The authors should consider adding all Supplementary Figures (S1-S4) to the main body of the manuscript for two reasons: not only they would help to generate more impact, but those figures are crucial parts of the study (even with this addition the manuscript would have only 8 figures in the text…).

Round 2

Reviewer 1 Report

I appreciate that the authors have addressed my comments and concerns in the revised version. The paper is now worthy of publishing in IJMS.